# Exploring the influence of historical storytelling on cultural heritage tourists' revisit intention: A case study of the Mogao Grottoes in Dunhuang

**Feng Yuxin**[1☉]**, Qin Jianpeng**[ORCID][1]*****, Lv Xiaoyu**[1☉]**, Tian Yunxia**[2☉]**, Meng Weilong**[3☉]

**1** College of Tourism, Northwest Normal University, Lanzhou, Gansu, China, **2** College of Economics and Management, Southeast University, Nanjing, Jiangsu, China, **3** College of Tourism, Hunan Normal University, Changsha, Hunan, China

☉ These authors contributed equally to this work.
* 3175612435@qq.com

**Data Availability Statement:** All relevant data are within the manuscript and its Supporting Information files.

## Abstract

The revisit intention of tourists has long been a focal point of academic inquiry. However, there is still insufficient research on the antecedents of revisit intention from the perspectives of historical storytelling, destination image and perceived value. Taking the Mogao Grottoes in Dunhuang, a UNESCO World Heritage Site, as a case study, this paper, based on stimulus–organism–response (SOR) theory, examines the impact of historical storytelling on the destination image, perceived value, and revisit intention. Additionally, it further explores the mediating role of destination image and perceived value, as well as the moderating effect of place attachment in this chain. The research findings indicate that: (1) Historical storytelling significantly enhances tourists' perception of the tourism experience and revisit intention; (2) The study supports the mediating effect of destination image and perceived value; (3) Place attachment has a significant positive moderating effect between historical storytelling and revisit intention. Effective historical storytelling can significantly enhance destination image and perceived value, improve tourists' participation and satisfaction in tourism, stimulate revisit intention, and promote the sustainable development of tourist destinations. These findings enrich the research content of cultural heritage tourism, providing valuable suggestions for improving the management level of cultural heritage tourism attractions and increasing visitors' revisit intention.

## Introduction

Cultural heritage tourism is an integral component of heritage tourism. As per the resolution of the 45th session of the World Heritage Committee, by October 2023, the total number of global heritage sites expanded to 1199, comprising 993 cultural heritage sites [1]. With the increasing internal demand for high-quality development of cultural tourism and the intensifying competition among similar tourist destinations, cultural heritage tourism destinations

**Funding:** The phased achievement of the project 'Research on the construction of iconic long march projects within Gansu province' funded by the Gansu Great Wall Long March National Cultural Park Construction and Development Research Center (001053108) awarded to FY. The 2023 graduate teaching case library construction project of Northwest Normal University: silk road China section cultural and tourism integration case library (2023YAL005) awarded to FY.

**Competing interests:** The authors have declared that no competing interests exist.

are also facing many challenges, especially in terms of how to maintain customer numbers and competitiveness [2]. Han & Hyun (2015) found that attracting repeat visitors can make destinations more cost-effective and beneficial for the sustainable development of cultural heritage sites [3]. However, current research on cultural heritage tourism mainly focuses on the protection and development of cultural heritage itself, and there is not enough comprehensive research on tourists' subjective behavioral intentions from the perspective of cultural heritage site visitors [4]. Therefore, it is necessary for us to study how to enhance the tourist perception and revisit intention of cultural heritage sites.

Historical storytelling, as a propaganda medium in the tourism process, can have a significant impact on tourists' participation and experience [5, 6]. Telling historical stories is considered a powerful tool to make historical events more attractive, while also effectively attracting tourists and enhancing their competitive advantage [7, 8]. Cultural heritage sites often have a long history and unique cultural heritage, providing important cultural scenes and ample historical materials for historical storytelling. Appropriate historical storytelling can help visitors better understand the development process of cultural heritage sites and the local distinctive culture, thus establishing an emotional connection between tourists and destinations, which is crucial for enhancing tourists' travel experience. Current research has applied historical storytelling to the tourism industry, exploring its impact on tourist emotional engagement and destination branding, and has found that it can indeed effectively enhance tourist perceptual experiences in different tourism settings [8, 9]. Leong, Yeh, Zhou, Hung, and Huan(2024) examined the impact of historical storytelling on tourists' perception of education, entertainment, experience, and emotional value in cultural heritage tourism sites through tour guide interaction and authentic place as intermediaries [6]. However, they failed to link these variables to travel outcomes, such as revisit intentions and word-of-mouth intentions. It is worth noting that the impact mechanism of historical storytelling on tourists' revisit intention has not been fully explored, requiring further theoretical exploration and empirical research.

Based on the SOR theory, this study aims to explore the influence of historical storytelling on tourists' revisit intentions in the context of cultural heritage tourism, in order to improve the quality of tourism services in scenic areas and enhance tourists' travel experiences, increase tourists' sense of participation and satisfaction, and stimulate tourists' revisit intentions. This study makes a significant contribution to the field of cultural heritage tourism. Firstly, starting from historical storytelling, a research model on the influence of historical storytelling on tourists' revisit intention was constructed, expanding the relevant research on tourists' revisit intention, enriching the research content of cultural heritage tourism, and providing alternative perspectives for the study of cultural heritage tourism. Secondly, this study examines the mediating effect of destination image and perceived value in the model, as well as the moderating effect of place attachment. This not only helps to theoretically understand how historical storytelling influences tourists' revisit intention, but also assists tourism destination managers in formulating appropriate development strategies, providing valuable insights for scholars and tourism practitioners.

## Literature review

### Stimulus–organism–response (SOR) model

The S-O-R model is derived from the behaviorist S-R model, incorporating the important role of internal states of individuals, abbreviated as the SOR model. The Stimulus-Organism-Response (SOR) theory was proposed by Mehrabian and Russell (1974), which includes stimulus, organism, and response, aiming to explore the role of external environmental stimuli in individual consumer behavior processes [10]. Stimulus refers to external factors that influence

individual cognition, organism refers to internal processes that intervene between stimuli and subsequent behaviors, and response is often conceptualized as individual approach or avoidance behaviors, including repurchase and recommendation, etc [11].

Initially applied in environmental psychology to study consumer purchasing behavior, the theory gradually expanded to investigate tourist consumption behavior, effectively explaining consumer behavior in most tourism contexts [12]. In recent years, a considerable number of scholars have embarked on research endeavors, leveraging this theory to investigate tourists' consumption behaviors and behavioral intentions. Liao et al. (2023), based on the SOR theory, analyzed the relationship between the types of reference group influence, perceived value, and health tourism intention, finding that both informational influence and utilitarian influence within reference group influence positively affect health tourism intention, with perceived value serving as a significant mediator between reference group influence and health tourism intention [13]. Therefore, this paper is based on the applicability of SOR theory, using the SOR framework as the theoretical foundation for the research.

In cultural heritage tourism research, scholars often focus on tourists' behavioral intentions and discuss their influencing factors [14]. However, historical storytelling as a factor influencing tourist behavioral intentions, has received less attention. Therefore, it is necessary to further study the impact of historical storytelling on the tourism experience and revisit intentions of tourists at cultural heritage sites. Based on this, applying the SOR model to this study, for cultural heritage tourism destination visitors, historical storytelling is established as "stimulus" and revisit intention is established as "response". Additionally, some scholars have found that destination image and perceived value can serve as mediating mechanisms behind the behavioral intentions of tourists at cultural heritage sites [15, 16]. Therefore, this mediating mechanism influencing the behavioral intentions of tourists at cultural heritage sites can be used (as an organism) to further validate the applicability of the SOR framework. Furthermore, some scholars have introduced place attachment as a moderating variable to explore its moderating effect on tourist behavioral intentions, and have found that place attachment has a significant moderating effect between tourist satisfaction and revisit intentions [17]. In summary, this study, based on the SOR theoretical model, considers historical storytelling as the "stimulus", destination image and tourist perceived value as the "organism", and tourist revisit intentions as the "response" to analyze the impact of historical storytelling on the revisit intentions of tourists at cultural heritage sites.

## Historical storytelling

Storytelling is a social phenomenon influenced by psychology, sociology, marketing, tourism, and behavioral sciences. Palombini (2017) defines storytelling as the narration of events that happen to certain individuals, leading to changes in the situation [18]. Building upon this, historical storytelling can be simply understood as the telling of historical events related to surrounding objects in appropriate settings. As scholars delve deeper into the study of historical storytelling, some have started to apply them in tourism research, aiming to enhance tourists' understanding of destinations, improve their perceived value, and stimulate their interest in travel [5]. Go and Govers (2012) found that storytelling greatly attracts tourists emotionally and cognitively, positively impacting their engagement and satisfaction [19]. In cultural heritage tourism, historical storytelling involves telling stories, anecdotes, and interesting facts related to cultural heritage vividly and engagingly. It is worth noting that Palombini (2017) emphasizes that storytelling in cultural heritage focuses on imparting specific knowledge about the historical events, cultural values, and social background of the depicted cultural heritage [18]. Against the backdrop of the flourishing development of cultural heritage tourism,

tourists' perception of the cultural heritage determines their cognitive image and value judgment of the destination, subsequently influencing their behavioral intentions. Historical storytelling serves as a crucial means to enhance tourists' cultural perception.

In recent years, scholarly interest in storytelling has proliferated across diverse disciplines, including politics, communication, cognitive science, anthropology, organizational behavior, and marketing, with a particular resonance in the domain of marketing [20]. Currently, the tourism industry and academia seem to widely acknowledge that storytelling is an effective communication tool that can attract potential tourists and promote and differentiate destinations [21]. Recent research endeavors have delved into various aspects of storytelling, encompassing theoretical frameworks, research methodologies, and the impact of storytelling on destination image and tourist experiences. Zhang and Ramyah (2024) conducted a comprehensive literature review on the role of storytelling in destination marketing, revealing that scholars predominantly employ Heider's Balance Theory and Narrative Transportation Theory in their investigations, with the latter gaining more traction in recent years [20]. Furthermore, it is observed that a majority of researchers favor qualitative approaches in studying storytelling, significantly outnumbering quantitative studies.

Furthermore, the examination of storytelling's influence on tourism destination image and visitor experience has consistently piqued the interest of researchers. For instance, Jo, Cha and Kim(2022) collected data from 259 tourists who experienced historical storytelling during their travels in South Korea, examining the influence of tourism storytelling on the tourism destination brand value, lovemarks, and relationship strength [22]. Gemar, Sánchez-Teba and Soler(2022) utilized survey-based methods and count models to assess tourists' perceptions of historical storytelling, destination image, and their intention to engage in cultural activities [23]. Furthermore, scholars have employed qualitative methodologies, such as interviews and observations, to explore the experiences and interpretations of tourists engaged with cultural heritage storytelling [2].

It is worth noting that recently some scholars have begun to explore the application of digital storytelling in the tourism industry. For example, Paiva et al. (2023) used a gaming experiment to find that digital storytelling can not only evoke people's perception of environmental degradation, stimulate ecological anxiety, sadness, and loneliness, but also connect these emotions with views on sustainable actions and future practices [24].

## Destination image

The destination image has a crucial role in influencing tourists' perception and opinions of the destination, playing a significant role in tourist behavior and travel purchase decisions [25]. Since the 1960s, the concept of image has been utilized in social and environmental psychology, consumer behavior research, and marketing, representing individuals' perception of objects, events, and behaviors driven by impressions, sensations, and beliefs [26]. When applied in the context of tourism, it is commonly referred to as "product brand image" or "destination image." In recent years, scholars have undertaken extensive research on tourist destination image, which can primarily be categorized into three aspects: the definition and dimensional constructs of destination image, the influencing factors of destination image, and its impact on tourists' behavioral intentions.

Most scholars have studied the definition of destination image from the perspective of individual tourists, but there is still no unified definition in current research. Crompton (1979) defines destination image as the sum of a person's trust, thoughts, and impressions of the destination [26]. Some scholars also consider destination image as a synthesis of people's perceptions, impressions, beliefs, and viewpoints of the destination [27]. In addition, the dimensional

division of destination image has always been a topic of interest to researchers, such as dividing it into single-dimensional, two-dimensional, three-dimensional, etc. Pike and Ryan (2004) believe that describing the destination image from the two dimensions of cognitive image and affective image can better capture the destination image. Cognitive image refers to tourists' objective cognition and views of the tourism destination, while affective image encompasses tourists' emotions towards the destination [28, 29].

Some scholars have discussed the factors influencing destination image. Balogu and McCleary (1999) categorized the influencing factors of destination image into personal factors and stimulus factors [30]. Building on this, Beelli and Martin (2004) expanded their research and summarized the influencing factors of destination image as individual factors and information sources, with personal factors typically including demographic variables, travel motivations, and destination familiarity; while information source factors are generally divided into first-hand information sources (experience and sensation) and second-hand information sources (social media and official websites, etc.) [31]. Of note, Wang et al. (2024) utilized innovation system theory and qualitative methods to explore the influencing factors and formation process of Cultural Inheritance-Based Innovation (CIBI) in Heritage Tourism Destinations (HTD), finding that CIBI is influenced by multilevel factors of the environment, government, enterprises, and public, encompassing basic innovation management conditions and sociocultural constraints [4].

Many scholars have also studied the impact of destination image on tourists' behavioral intentions, dividing it into three stages: pre-travel, during travel, and post-travel. We find that most scholars primarily focus on the influence of destination image on tourists' post-travel behavioral intentions [15]. Gavurova et al. (2023) investigate the impact of tourism destination image on travelers' behavioral intentions from a transportation perspective, utilizing cluster analysis and regression analysis to quantify the relationship between road transport development indicators and visitor spending, revealing that traffic conditions significantly influence visitors' willingness to pay [32]. Yang et al. (2024) explored the heterogeneous effects of different tourism advertising styles on cultural heritage sites. The study found that compared to direct tourism advertising, metaphorical tourism advertising can shape a more attractive image of the tourist destination, thus stimulating stronger visitation intentions among cultural heritage site visitors [14].

## Perceived value

Perceived value is an important concept and theory in researching consumer experience and experience quality. The concept of perceived value was first proposed by American service marketing scholar Zeithaml (1988) in the field of marketing. He believes that perceived value is the overall feeling that customers have after consuming a certain product or service, evaluating the input and output [33]. In recent years, scholars have studied various aspects of tourist perceived value, including the concept of perceived value, dimensions of perceived value, and its impact on tourist behavioral intentions, continuously expanding and improving the theory of tourism perceived value.

Due to the different understandings of perceived value among scholars, there is diversity in the definition of perceived value. Its definition can mainly be analyzed and discussed from perspectives such as cost and benefit, input and output ratio [34, 35]. Sáncheztal (2006) suggests that the concept of perceived value can be regarded as a multidimensional structure, which has been widely applied in academia [36]. This approach enables us to overcome some of the issues associated with understanding perceived value using traditional methods, particularly the excessive focus on economic utility [33]. Furthermore, Scholars have divided perceived value

into different dimensions according to their research needs, such as single-dimensional, two-dimensional, and three-dimensional, without a clearly unified method for dimensional division. For example, the two-dimensional scale proposed by Grewal et al. (1998) is highly representative [37]. They divided the dimensions of perceived value from the perspective of transaction processes into transaction value and acquisition value, which is considered a reliable and effective measurement method [38].

Tourist perceived value is the subjective evaluation made by tourists regarding the services and products obtained during the travel process, which can exert a positive effect on aspects such as tourist satisfaction and behavioral intentions [39, 40]. Furthermore, it also influences both the pre-purchase and post-purchase stages. Empirical research conducted by many scholars has demonstrated that high levels of tourist perceived value can positively affect their revisit intention. For example, Peng et al. (2023) explored the impact mechanism of tourists' perceived happiness on the revisit intention to traditional Chinese medicine cultural tourism destinations based on attachment theory [2]. Through empirical research, it was found that when tourists perceive a higher level of happiness at cultural heritage sites, their revisit intention is significantly enhanced. Yao et al. (2020) took the case of Meizhou Island, the birthplace of China's first world-class intangible cultural heritage—Mazu belief, to explore the significant positive impact of tourist perceived value on revisit intention, with place attachment playing a partial mediating role [40].

## Revisit intention

Tourist revisit intention is one of the key indicators for sustainable development in the tourism industry. Firstly, compared to first-time visitors, repeat visitors tend to have a greater economic contribution to the tourism destination because of their increased interest in the destination, leading them to invest more time and effort, as well as purchase more tourism products and services, thereby extending the life cycle of the tourism destination to some extent [41, 42]. Secondly, from the perspective of tourist consumption behavior, retaining an existing customer is much cheaper than acquiring a new one [43]. Consequently, scholars have conducted extensive research on how to enhance tourist loyalty and stimulate revisit intention, categorizing tourist loyalty into attitudinal loyalty and behavioral loyalty, with revisit intention being a critical manifestation of tourist behavioral loyalty [44, 45].

Tourists' revisit intention reflects the intention that tourists may want to visit a tourist place again, and this revisit intention can be transformed into actual revisit behavior under certain conditions [44]. In terms of research content, early studies identified factors influencing revisit intentions primarily as tourists' income, age, and occupation, among other external conditions. Chen and Huang (2010), based on network text analysis and in-depth interviews, empirically researched the motives for revisiting among tourists in Xiamen city [46]. They concluded that factors such as age, occupation, and monthly income significantly influence tourists' revisit intention. It is worth noting that many early scholars overlooked internal factors such as psychological processes during tourists' travel experiences. Subsequently, scholars gradually recognized the importance of psychological factors such as travel motivation, satisfaction, destination image, and travel involvement in influencing tourists' revisit intention. In current research on tourists' revisit intention, scholars primarily focus on aspects such as motivation [47], satisfaction [48], destination image [15], and perceived value [40], among others. It has been found that these factors positively promote tourists' revisit intention.

In addition, in terms of research methodology, scholars commonly employ traditional quantitative analysis methods to investigate tourists' revisit intentions, such as Structural Equation Modeling and Logistic Regression analysis [2, 49]. However, these models assume

that the variables do not affect each other, which may lead to omissions and deficiencies in the research.

In general, scholars have extensively studied the factors influencing tourists' revisit intention, and it has been found that the revisit intention is the result of a cumulative effect based on multiple factors, necessitating consideration of various influences. The influence of external characteristics tends to be secondary in comparison to the role of emotional factors. Therefore, it is necessary to prioritize the perceptions and tourism experiences of visitors, conducting in-depth analyses of their practical needs and psychological motivations, in order to effectively enhance their revisit intention. For example, Rasoolimanesh et al. (2021) used the case study of the world heritage city of Kashan in Iran to empirically investigate how visitor participation, authenticity, and destination image directly or indirectly affect heritage site visitors' revisit intention and electronic word-of-mouth (eWOM) intention through memorable tourism experiences (MTE) [50].

## Place attachment

In the 1970s, geographer Tuan (1974) first proposed the concept of "topophilia" to summarize the emotional connection between individuals and specific material spaces, including perceptions, values, and worldviews [51]. He conducted preliminary research on this concept, which laid the theoretical foundation for place attachment. Originating from environmental psychology, the theory of place attachment gradually became an important concept in disciplines such as geography, tourism studies, and urban and rural planning, gaining widespread attention. As research on place attachment deepened, many scholars applied it to tourism studies, analyzing and exploring the emotional connections between people and places by treating place attachment as an independent variable, a mediating variable, or a moderating variable. Dang and Weiss (2021) conducted a systematic literature review examining the relationship between place attachment and behavioral intentions [52]. Through qualitative analysis, they found that many scholars have empirically demonstrated the significant impact of place attachment on factors such as willingness to pay, loyalty, risk coping behavior, pro-environmental behavior, and pro-tourism behavior.

In recent years, research on place attachment in the tourism industry has mainly focused on dimensions and their impact on visitor satisfaction, loyalty, and behavioral intentions. Most scholars believe that place attachment is a multidimensional concept, thus various methods of dimension division research have been proposed, such as single-dimensional, two-dimensional, and three-dimensional, etc., but there is still no clear-cut division method. However, it is worth noting that Williams et al. (1992) proposed a division of place attachment into two dimensions: place dependence and place identity. Place identity emphasizes emotional connection, while place dependence leans towards people's reliance on the functional aspects of a place. This two-dimensional model has been validated and acknowledged by the academic community [53].

In addition, many tourism scholars have found that place attachment has a direct or indirect positive relationship with tourists' satisfaction, loyalty, and behavioral intentions. This study focuses on the impact of place attachment on tourists' revisit intention. Place attachment refers to the special emotions that people have towards a place due to various factors, and this special emotion is an important influencing factor in forming tourists' revisit intention, playing a facilitating role in their decision-making process [54]. At the same time, cultural heritage sites are more likely to evoke special emotions in tourists due to their unique historical, cultural, and emotional values, thereby fostering a sense of place attachment and stimulating revisit intentions [2, 40].

## Hypotheses development

In conclusion, scholars have been conducting in-depth research on historical storytelling and revisit intentions, and have made certain research achievements, providing a theoretical basis and guidance for the development of this study.

Storytelling is ubiquitous in every aspect of life, as it can help people connect with each other and cultivate empathy, leading individuals to attempt to "feel the world in the same way as the characters in the story" [55, 56]. In the digital age, historical storytelling is considered an effective means to enhance the reputation of destinations [57]. When tourism service personnel tell historical stories at cultural heritage sites, they create immersive and memorable experiences for tourists, connecting them with the local social and cultural context, providing higher quality tourism services, and helping tourists better understand the destination, stimulating their interest in tourism, and increasing their participation and satisfaction [6]. Revisit intention usually refers to the likelihood of tourists returning to the destination, with factors influencing revisit intention mainly including satisfaction, destination image, perceived value, etc. [58]. Although the presence of historical storytelling is not among the numerous influencing factors, existing research has found that historical storytelling has a positive impact on the tourist experience and perceived value of cultural heritage sites [6]. Therefore, this study believes that telling historical stories at cultural heritage sites can provide visitors with a better tourism experience, enhance visitors' cognitive image, perceived value, and revisit intention. Thus, this study proposes the following hypotheses:

H1: Historical storytelling positively influences the destination image.

H2: Historical storytelling positively influences tourists' perceived value.

H3: Historical storytelling positively influences tourists' revisit intention.

The image of a tourism destination reflects tourists' overall perception and impression of the destination, and a good destination image can effectively stimulate tourists' willingness to visit and influence their travel decisions [15]. Zhang and Hou (2014) conducted an empirical study using the case of Hongcun, a World Cultural Heritage site in Anhui province, which elucidated that tourists' positive perceptions of a destination's image significantly encourage revisit intention [59]. Furthermore, Lai et al. (2020) used Australia as an example to predict the behavioral intentions of potential Chinese tourists by combining cognitive and affective components of food imagery. The results showed that cognitive food imagery was more predictive of tourist behavioral intentions than affective imagery [60]. Therefore, this study focuses primarily on the impact of destination cognitive imagery on tourists' travel experiences and behavioral intentions.

In addition, many scholars have recently studied the relationship between destination image and tourist perceived value. Through empirical research, they have found that destination image has a significant impact on perceived value, with destinations having a favorable image often exhibiting higher perceived value [61]. Gu and Cui (2023) used Hangzhou, which possesses three world cultural heritage sites, as a case study and found that tourists' cognition of world cultural heritage significantly influences their perceived quality and perceived value [62]. Therefore, this study proposes the following hypothesis:

H4: The image of a tourist destination positively influences tourists' revisit intention.

H5: The destination image of tourism positively influences tourists' perceived value.

Based on the aforementioned literature review, we discern a positive influence of tourists' perceived value on their revisit intentions, an effect that persists within the context of cultural

heritage tourism [2]. For example, Han (2015) takes the Ling Shan Buddhist cultural leisure resort in Wuxi as an example, and through empirical research, it is found that perceived value is the antecedent variable of satisfaction and behavioral intention, and the perceived value of cultural tourist destinations will directly affect their behavioral willingness [63]. Therefore, this study proposes the following hypotheses:

H6: Tourist perceived value positively influences tourist revisit intention.

Based on the previous assumptions, this study attempts to delve into the potential chain-mediated effects in the research model. Existing literature has analyzed the relationship between destination image, perceived value, and tourists' revisit intention, revealing that perceived value plays a mediating role between destination image and tourists' revisit intention [64]. Meanwhile, many scholars tend to focus only on the overall perceived value when exploring the mediating role of perceived value between destination image and behavioral intention [16, 40]. Therefore, this study mainly focuses on the role of overall perceived value in the model.

Xu and Li (2018) studied the mechanism of the effect of destination image in Xinjiang, China on tourists' behavioral intentions, finding that perceived value plays a crucial role in the impact mechanism of destination image perception on tourists' behavioral intentions [16]. However, this relationship has not yet been fully explored in the specific context of cultural heritage tourism. This study argues that historical storytelling can effectively enhance tourists' perception and views of a destination, thereby enabling tourists to experience higher value and ultimately strengthening their revisit intention. Therefore, this study considers historical storytelling as the antecedent variable, destination image and perceived value as the mediating variables, and revisit intention as the outcome variable, to delve into the potential chain-mediated effects. Thus, this study proposes the following hypotheses:

H7: The destination image plays an intermediary role between historical storytelling and tourists' revisit intention.

H8: Tourist perceived value plays an intermediary role between historical storytelling and tourists' revisit intention.

H9: The destination image and tourist perceived value play a mediating role in the relationship between historical storytelling and tourists' revisit intention.

Furthermore, based on the literature review above, we can find that place attachment has a significant impact on the revisit intention [63]. It is worth noting that Yun and Liu (2023) introduced place attachment as a moderating factor into the research framework, and found that place attachment can moderate the impact of perceived distance on tourists' behavioral intentions [65]. Furthermore, they discovered that perceived value mediates the moderating effect of place attachment on tourists.

Place attachment is the connection formed between individuals and places through interaction, which can influence tourists' emotions and behaviors towards tourism destinations [52]. In cultural heritage tourism, this attachment may become more profound and intricate due to the unique character and historical significance of the cultural heritage sites [2]. Therefore, this study believes that different levels of place attachment will moderate tourists' revisit intention. This study suggests that when tourists have a high level of place attachment to the destination, historical storytelling will make it easier for them to develop a desire to revisit, and the mediating effect of destination image and perceived value between historical storytelling and revisit intention will also be strengthened. Therefore, this study proposes the following hypothesis:

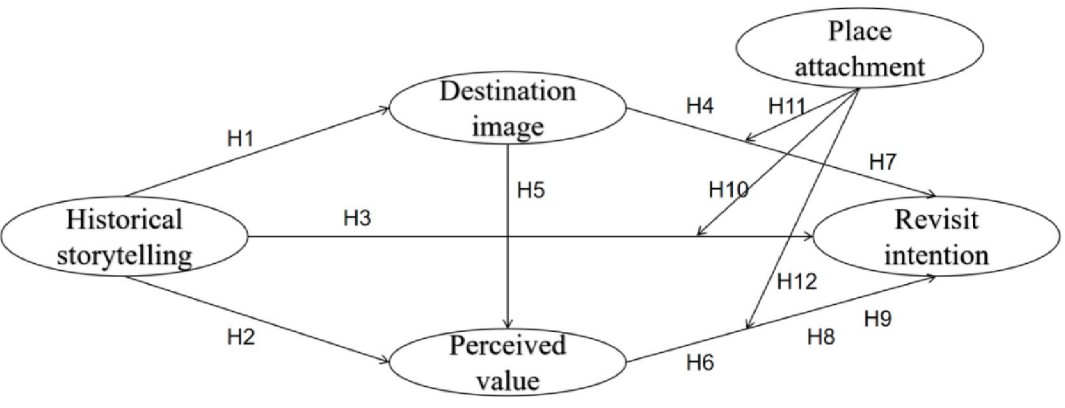

**Fig 1. Research model.**

H10: Place attachment plays a positive moderating role in the impact of historical storytelling on the revisit intention.

H11: Place attachment positively moderates the mediating role of destination image between historical storytelling and revisit intention in tourism.

H12: Place attachment positively moderates the mediating role of perceived value in historical storytelling and revisit intentions for tourists.

In conclusion, based on the above analysis and reasoning, this study has constructed a research model from the perspective of the SOR framework, which includes historical storytelling, destination image, perceived value, place attachment, and revisit intention (Fig 1). Furthermore, in this research model, destination image and perceived value play a mediating role, while place attachment has a moderating effect.

## Methodology

### Research scope and object

This study takes the Mogao Grottoes in Dunhuang, a UNESCO World Heritage site, as a case study. Popularly known as the Thousand Buddha Caves, the Mogao Grottoes are hailed as the "most valuable cultural discovery of the 20th century" and the "Eastern Louvre". They are the largest and most abundant Buddhist art site in the world, possessing significant historical and cultural value as well as tourism value. Meanwhile, the Dunhuang Mogao Grottoes serve as a microcosm of the evolution of Chinese cave art, representing the longest-lasting, most rich in content, artistically exquisite, well-preserved, and most influential grotto group in China [66]. In 1987, the Mogao Grottoes were designated as one of the first World Cultural Heritage sites in China, enjoying high international recognition and attracting a large number of visitors for sightseeing. This can provide a representative sample for empirical research on the tourist experience and revisit intentions of cultural heritage sites.

In Dunhuang, where the tourism industry accounts for a significant proportion of the tertiary sector at 63.33%, the Mogao Grottoes undoubtedly serve as the cornerstone of its tourism brand. The development of Mogao Grottoes tourism directly determines the prosperity or decline of Dunhuang's tourism economy. According to statistics from the cultural and tourism department of Dunhuang City, in 2023, the Dunhuang tourism industry, centered around the Mogao Grottoes, received a total of 16.82 million visitors and achieved tourism revenue of 15.6 billion yuan, both reaching historical highs [67]. In recent years, the Dunhuang Research

Institute has further developed cultural experiential projects to enhance the Dunhuang tourism experience and attract more visitors. For example, the digital Dunhuang project "Seeking Dunhuang" combines advanced game technologies such as 3D modeling and VR virtual reality scenes to lead visitors to "travel through" historical time and "explore" the thousand-year-old Dunhuang. Compared with other cultural heritage sites, the Mogao Grottoes have experienced thousands of years of history, where various cultures from ancient and modern times, both domestic and foreign, converge and blend. This long and ancient history provides an important cultural backdrop and ample historical materials for historical storytelling, as well as facilitates our study of the impact of historical storytelling on tourists' travel experiences and revisit intentions.

## Construct measurement and questionnaires

This study employed a questionnaire survey comprising two parts. The first part aimed to collect respondents' basic information. The second part aimed to measure levels of various constructs. Historical storytelling consisted of four items aimed at assessing the persuasive level of local historical storytelling in Mogao Grottoes [6, 68]. Destination image included seven items aimed at gauging tourists' perceptions of various aspects of the tourist destination [69, 70]. Perceived value consisted of three items aimed at evaluating the overall assessment of tourists on the tourism process [48, 71]. Place attachment comprised six items aimed at measuring the emotional connection between people and places [53]. Revisit intentions by tourists included three items aimed at assessing tourists' intention to revisit the Mogao Grottoes scenic area [2, 72]. These items were formatted using a 7-point Likert scale, where 1 indicated low levels and 7 indicated high levels. The questionnaire's five latent variables totaled 23 items. Considering the quality and effectiveness of the questionnaire, investigators conducted a survey in advance on 51 tourists who had experienced historical storytelling at the Mogao Grottoes in Dunhuang. Based on the analysis results, one item with a low factor loading in the destination image was removed, resulting in the formal survey questionnaire on the intention of revisiting the Mogao Grottoes. Please refer to the S1 Appendix for the questionnaire.

## Data collection and analysis

This study employs a convenience sampling method to conduct a survey on visitors at the Mogao Grottoes scenic area. This non-probability sampling strategy is deemed cost-effective and suitable for an on-site tourist survey in behavioral science research [73]. While the target population and survey locations were intentionally chosen, the sampling process was spontaneous, which will reduce research bias [74]. The data collection for this questionnaire is mainly based on the self-completion method. This study obtained data from two channels, online and offline. Firstly, the researchers distributed survey questionnaires to local tourists randomly at the Mogao Grottoes in Dunhuang during the peak tourist season. Data collection was conducted during two time periods to ensure the diversity of the data, from October 1, 2023 to October 3, 2023, and from January 1, 2024 to January 3, 2024. A total of 244 questionnaires were distributed and all were collected. Secondly, from February 3, 2024 to March 3, 2024, researchers distributed online survey questionnaires through the Wenjuanxing platform and received a total of 107 completed questionnaires. After excluding 26 incomplete questionnaires, a total of 325 valid surveys were obtained, representing an effective response rate of 92.6%. There is no consensus among scholars regarding the selection of sample size. Comrey (1978) suggests that the minimum sample size should be at least 200, while Nunaly (1978) suggests that the sample size should be at least ten times the number of variables [75, 76]. Therefore, the 325 samples in this study meet the requirements of the above scholars, ensuring the

accuracy and reliability of the model. This study used statistical software SPSS 26.0 and AMOS 26.0 to analyze the data from a total of 325 questionnaires collected online and offline.

## Results

### Respondent profile

In the 325 samples, there were 138 males and 187 females. Among the respondents, 228 were aged between 26 and 40, while 65 were aged between 41 and 60. In terms of education, 261 respondents had a bachelor's degree, and 46 had a master's degree or higher. Regarding monthly income, 164 respondents had a monthly income between 5000 and 10000 RMB, while 82 had a monthly income between 10001 and 20000 RMB. It can be observed that visitors to the Mogao Grottoes generally have higher incomes. Additionally, only 102 respondents had visited the Mogao Grottoes before, accounting for 31.4% of the total. Therefore, research on visitors' revisit intention is crucial for the future development of Mogao Grottoes tourism. For detailed information on the respondents, please refer to Table 1.

### Data reliability and validity

By examining the specific indicators of each item in the model, the reliability and validity of the model can be measured. First, the reliability of the model can be assessed by examining the value of Cronbach's Alpha. Nunnally (1978) suggested that a Cronbach's Alpha value greater than 0.7 indicates acceptable reliability, while a value greater than 0.8 indicates good reliability [76]. As shown in Table 2, the Alpha values of each latent variable range from 0.778 to 0.936, indicating relatively good reliability of the scale. Secondly, the convergent validity of the model can be evaluated by examining the factor loading, composite reliability (CR), and average variance extracted (AVE) of each latent variable. According to the standards proposed by Fornell and Larcker (1981), a CR value greater than 0.6 and an AVE value greater than 0.5 indicate an acceptable level of convergent validity [77]. Pearce and Lee (2005) suggested that in addition to

**Table 1. Demographics descriptive data analysis.** (n = 325).

| Variable | n | % | Variable | n | % |
|---|---|---|---|---|---|
| **Gender** | | | **Occupation** | | |
| Male | 138 | 42.5 | Government employees | 35 | 10.8 |
| Female | 187 | 57.5 | Sole trader | 74 | 22.8 |
| | | | Enterprise employee | 92 | 28.3 |
| **Age** | | | Student | 21 | 6.5 |
| Under 18 | 4 | 1.2 | Retired | 31 | 9.5 |
| 18–25 | 25 | 7.7 | Other | 72 | 22.2 |
| 26–40 | 228 | 70.2 | | | |
| 41–60 | 65 | 20 | **Education** | | |
| Over 60 | 3 | 0.9 | Junior high school and below | 1 | 0.3 |
| | | | Senior high school | 17 | 5.2 |
| **Monthly income** | | | College | 261 | 80.3 |
| 5,000 yuan or less | 42 | 12.9 | Master's degree and above | 46 | 14.2 |
| 5,001–10,000 yuan | 164 | 50.5 | | | |
| 10,001–20,000 yuan | 82 | 25.2 | **Have you ever visited the Mogao Caves before?** | | |
| 20,001–30,000 yuan | 33 | 10.2 | Yes | 102 | 31.4 |
| 30,001 yuan or more | 4 | 1.2 | No | 223 | 68.6 |

**Table 2. Results of reliability and convergent validity testing.**

| Construct measures | Factor Loading | Cronbach's α | AVE | CR |
|---|---|---|---|---|
| **Historical storytelling** | | 0.936 | 0.791 | 0.938 |
| The story of the place is engaging | 0.937 | | | |
| The story of the place is very memorable | 0.908 | | | |
| The story helps me to understand the history of the place | 0.913 | | | |
| The story helps me to understand the people in the past | 0.798 | | | |
| **Destination image** | | 0.882 | 0.532 | 0.872 |
| This place offers suitable accommodation | 0.742 | | | |
| This place has high-quality infrastructure | 0.757 | | | |
| The place offers attractive local food | 0.740 | | | |
| This place has a standard of hygiene and cleanliness | 0.716 | | | |
| The inhabitants of this place are interesting and friendly | 0.779 | | | |
| This place has beautiful scenery | 0.737 | | | |
| **Perceived value** | | 0.778 | 0.601 | 0.817 |
| Overall, I think the value of my travel experience is high | 0.896 | | | |
| Compared to the time and energy I paid, I think I have received good value | 0.747 | | | |
| Compared to the price I paid, I think I have received good value | 0.666 | | | |
| **Revisit intention** | | 0.851 | 0.709 | 0.878 |
| I plan to visit the Mogao Grottoes again in the future | 0.913 | | | |
| I will probably travel to the Mogao Grottoes again in the next five years | 0.680 | | | |
| Given the opportunity, I will come to the Mogao Grottoes next time | 0.908 | | | |
| **Place attachment** | | 0.874 | 0.536 | 0.874 |
| The Mogao Grottoes provided me with a unique tourism experience. | 0.744 | | | |
| Compared to other tourist destinations, the tourism facilities and offerings at the Mogao Caves better cater to my needs. | 0.686 | | | |
| Compared to other tourist destinations, I prefer Mogao Grottoes. | 0.732 | | | |
| There is a strong sense of identification with the Mogao Grottoes. | 0.757 | | | |
| I have a deep affection for the Mogao Caves. | 0.723 | | | |
| Visiting the Mogao Caves is very meaningful to me. | 0.750 | | | |

Note: CR = composite reliability; AVE = average variance extracted.

meeting the former criteria, the standardized factor loading of all measurement items should be greater than 0.6 to meet the standard [78]. As shown in Table 2, the factor loading, CR, and AVE values of each latent variable all meet the standards, indicating good convergent validity of the scale.

In addition, discriminant validity testing of all latent variables can evaluate the discriminant validity of the model. According to the standards proposed by Fornell and Larcker (1981), the square root of the average variance extracted (AVE) should be greater than the inter-variable correlations [77]. As shown in Table 3 below, the square root of the AVE for each latent

**Table 3. Results of the discriminant validity test.**

| Construct | Place attachment | Historical storytelling | Destination image | Perceived value | Revisit intention |
|---|---|---|---|---|---|
| **Place attachment** | **0.732** | | | | |
| **Historical storytelling** | 0.589 | **0.891** | | | |
| **Destination image** | 0.416 | 0.707 | **0.746** | | |
| **Perceived value** | 0.421 | 0.715 | 0.663 | **0.775** | |
| **Revisit intention** | 0.502 | 0.717 | 0.677 | 0.699 | **0.841** |

**Table 4. Model fitting result.**

| Model Indexes | CMIN/DF | GFI | NFI | RFI | IFI | TLI | CFI | RMSEA |
|---|---|---|---|---|---|---|---|---|
| Value | 1.711 | 0.913 | 0.930 | 0.919 | 0.970 | 0.965 | 0.969 | 0.047 |
| Suggested Values | 1–3 | >0.9 | >0.9 | >0.9 | >0.9 | >0.9 | >0.9 | <0.05 |

variable on the diagonal is greater than the correlations between the corresponding variables, indicating good discriminant validity of the scale.

## Structural model and hypotheses test

**Main effect analysis.** This study used AMOS 26.0 to test the model fit and path relationships. As shown in Table 4, according to the standards proposed by Wu (2009), CMIN/DF (chi-square degrees of freedom ratio) = 1.711, within the range of 1–3, RMSEA (root mean square error of approximation) = 0.047, within the excellent range of <0.05 [79]. Additionally, the test results of GFI, NFI, and RFI indicators all reached excellent levels of above 0.9. Therefore, the results of model fit testing indicate that the model in this study has a good fit.

Next, as shown in Table 5, historical storytelling has a significant positive impact on the destination image (β = 0.707, p<0.001), perceived value (β = 0.493, p<0.001), and revisit intention (β = 0.254, p<0.001). Additionally, the destination image has a significant positive impact on the perceived value (β = 0.315, p<0.001) and revisit intention (β = 0.246, p<0.001). Finally, the perceived value has a significant positive impact on the revisit intention (β = 0.303, p<0.001). Therefore, this study assumes that hypotheses H1-H6 are all valid.

**Mediation effect analysis.** This study further explores the chained mediation effect of destination image and revisit intention in the model. In testing the mediation effect, MacKinnon et al. (2004) argue that the bootstrapping method is superior to the causal steps test and the Sobel test [80]. According to Hayes (2009), if the confidence interval between the upper and lower limits does not include zero, then the mediation effect is significant [81]. As shown in Table 6, firstly, destination image significantly mediates the relationship between historical storytelling and revisit intention (β = 0.174, p<0.05); secondly, perceived value significantly mediates the relationship between historical storytelling and revisit intention (β = 0.149, p<0.05); finally, historical storytelling significantly influences revisit intention through the chained mediation of destination image and perceived value (β = 0.067, p<0.05). Therefore, this study assumes that H7-H9 are all supported.

**Moderating effect analysis.** According to Edwards and Lambert (2007), the regression coefficients were tested using model 89 from the PROCESS 4.1 plugin in SPSS 26.0. The interaction effects between the moderator variable and the mediator variable were examined using the bootstrapping method, and the changes in the mediated effect after moderation were

**Table 5. Results of model path relationship test.**

| Path | | | Estimate | S.E. | C.R. | P | $R^2$ |
|---|---|---|---|---|---|---|---|
| Destination image | <— | Historical storytelling | 0.707 | 0.053 | 11.849 | *** | 0.500 |
| Perceived value | <— | Historical storytelling | 0.493 | 0.053 | 6.38 | *** | 0.561 |
| Perceived value | <— | Destination image | 0.315 | 0.059 | 4.106 | *** | |
| Revisit intention | <— | Historical storytelling | 0.254 | 0.053 | 3.339 | *** | 0.622 |
| Revisit intention | <— | Destination image | 0.246 | 0.057 | 3.401 | *** | |
| Revisit intention | <— | Perceived value | 0.303 | 0.077 | 3.999 | *** | |

Note: *** *p* < 0.001.

**Table 6. Results of standardized mediation effect tests.**

| Paths | Estimate | SE | Bias-corrected 95%CI | | | Percenntile 95%CI | | |
|---|---|---|---|---|---|---|---|---|
| | | | Lower | Upper | P | Lower | Upper | P |
| stdIndA1 | 0.174 | 0.050 | 0.074 | 0.270 | 0.001** | 0.078 | 0.275 | 0.000*** |
| stdIndA2 | 0.149 | 0.049 | 0.069 | 0.260 | 0.000*** | 0.067 | 0.253 | 0.001** |
| stdIndA3 | 0.067 | 0.025 | 0.030 | 0.131 | 0.000*** | 0.026 | 0.123 | 0.001** |

Note: Standardized estimating of 5000 bootstrap samples

**$p < 0.01$

***$p < 0.001$. stdIndA1:Historical storytelling→Destination image→Revisit intention; stdIndA2:Historical storytelling→Perceived value→Revisit intention; stdIndA3: Historical storytelling→Destination image→Perceived value→Revisit intention.

ultimately determined [82]. As shown in Table 7, after controlling for age, education, monthly income, and visit frequency, the study found that place attachment positively moderated the relationship between historical storytelling and revisit intention (b = 0.171, p<0.05). The interaction term of destination image and place attachment (b = 0.046, p>0.05) and the interaction term of perceived value and place attachment (b = -0.156, p>0.05) do not have a significant impact on revisit intention. Therefore, H10 was supported, while H11 and H12 were not supported. Furthermore, the previous visit to the destination had a significant impact on tourists' revisit intention.

In order to further analyze the moderating effect of different levels of place attachment on the relationship between historical storytelling and revisit intentions, this study conducted a simple slope analysis. As shown in Fig 2, when there is a high sense of place attachment (b = 0.256, p<0.01), the positive impact of historical storytelling on revisit intentions is significant, while when there is a low sense of place attachment (b = 0.068, p>0.05), the positive effect is not significant. This finding further supports H10.

**Table 7. Results of moderated mediation analysis.**

| | Revisit intention | | |
|---|---|---|---|
| | coeff | se | t |
| constant | 6.153 | 0.172 | 35.847*** |
| Historical storytelling | 0.162 | 0.042 | 3.892*** |
| Destination image | 0.219 | 0.047 | 4.651*** |
| Perceived value | 0.282 | 0.046 | 6.058*** |
| Place attachment | 0.103 | 0.042 | 2.455* |
| Historical storytelling x Place attachment | 0.171 | 0.075 | 2.291* |
| Destination image x Place attachment | 0.046 | 0.075 | 0.611 |
| Perceived value x Place attachment | -0.165 | 0.103 | -1.598 |
| Age | -0.032 | 0.034 | -0.95 |
| Education | -0.03 | 0.042 | -0.697 |
| Monthly income | 0.007 | 0.023 | 0.312 |
| Have you ever visited the Mogao Caves before? | -0.113 | 0.04 | -2.846* |
| R-sq | 0.563 | | |
| F | 36.725 | | |

Note: Standardized estimating of 5000 bootstrap samples

*$p < 0.05$

**$p < 0.01$

***$p < 0.001$.

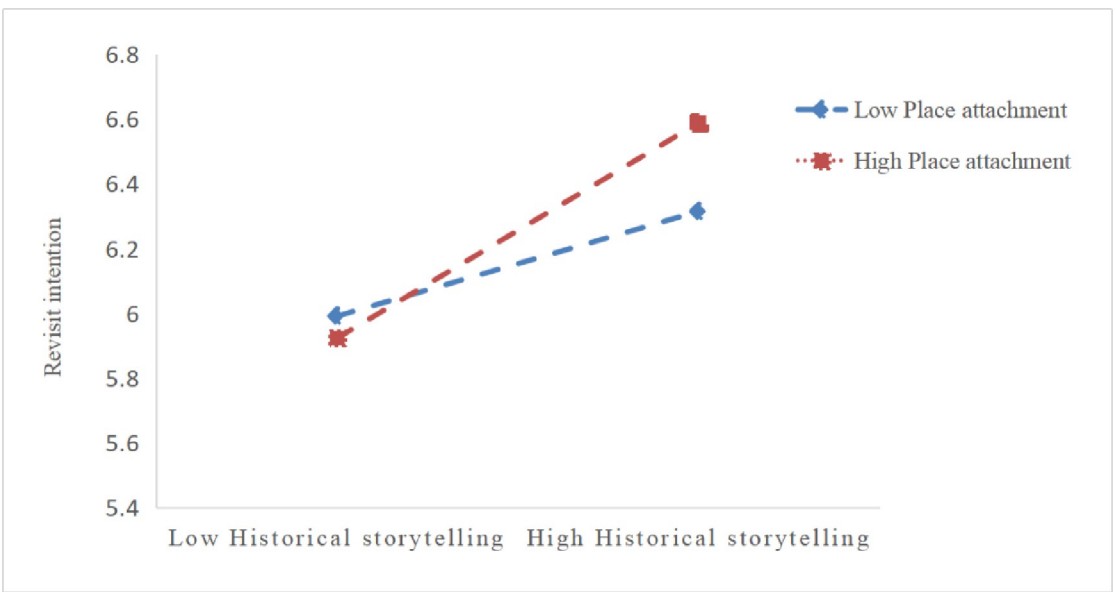

**Fig 2. The moderating role of place attachment between historical storytelling and revisit intention.**

## Conclusion and prospects

### Conclusion and discussion

This study explores the chain mediation effect between historical storytelling and revisit intention through the mediating variables of destination image and perceived value. Building on this, further analysis was conducted on the moderating effect of place attachment on revisit intentions across different paths. In conclusion, the results of this study provide valuable insights into the relationships among historical storytelling, destination image, perceived value, place attachment, and revisit intention.

The main findings can be summarized into three key points. Firstly, the establishment of hypotheses H1, H2, and H3 indicates that historical storytelling can enhance the destination's cognitive image and perceived value, stimulating tourists' revisit intention. This is consistent with the research findings of Go and Govers (2012), who suggest that effective historical storytelling has a positive impact on tourist engagement and satisfaction [19]. Our research further demonstrates that even in the absence of explicit destination image and perceived value, historical storytelling itself can have a positive impact on revisit intentions. However, the lack of these intermediate variables would not be conducive to fully stimulating tourists' revisit intentions. The establishment of hypotheses H7, H8, and H9 suggests that the chain-mediated effect of destination image and perceived value is beneficial for historical storytelling to better stimulate tourists' revisit intention. This means that by shaping positive destination images and enhancing perceived value, historical storytelling can more effectively stimulate tourists' revisit intention.

Secondly, this study explores the relationship between destination image, perceived value, and revisit intention. Based on the hypotheses H4, H5, and H6, the study found that a better destination cognitive image is conducive to enhancing tourists' perceived value. The destination's cognitive image and perceived value will have a significant positive impact on tourists' revisit intention, which is consistent with the findings of Xu and Li. Xu and Li (2018) found that perceived value plays a crucial role in the impact of destination cognitive image on tourists' behavioral intentions [16]. Through empirical research using this model, it was found that

this relationship also holds true in cultural heritage tourism. Furthermore, this study found that compared to destination image, perceived value has a greater impact on revisit intention.

Finally, according to the established hypothesis H10, place attachment plays a significant positive moderating role between historical storytelling and revisit intention. This study was conducted in the context of cultural heritage tourism, revealing that the impact of historical storytelling on revisit intention varies across different levels of place attachment. When tourists have a higher level of place attachment, the influence of historical storytelling on their revisit intention is strengthened. The non-confirmation of Hypotheses H11 and H12 suggests that place attachment does not exert a significant moderating influence on the mediating effect of destination cognitive image and perceived value in the relationship between historical storytelling and revisit intention. This indicates that in cultural heritage tourism, place attachment is an important factor influencing tourists' revisit intention, but its role is mainly reflected in directly enhancing the attractiveness of historical storytelling to tourists, rather than indirectly affecting tourists' revisit intention by adjusting other variables (such as destination cognitive image and perceived value). This is also similar to the research results of Hu, Wang, and Yun et al., suggesting that place attachment may have a certain moderating and promoting effect on revisit intention [17, 65, 83]. Wang et al. (2022) found that by introducing place attachment as a moderating variable, place attachment plays a positive moderating role in the relationship between pro-environmental behavior and revisit intention [83].

## Theoretical implications

This study constructs a research model of tourists' revisit intention based on the SOR theory, enriching and broadening the research on tourists' revisit intention, and has certain theoretical significance. The theoretical significance of this study mainly includes the following three points.

Firstly, the construction of the model for revisiting intentions of tourists at cultural heritage tourism sites in this study contributes to the refinement of the management theory of cultural heritage tourism. Kim (2014) argued that research on tourist experiences should shift the focus from the objective attributes of destinations to the subjective interpretations of these attributes by tourists [84]. Our study extends the research on tourists' revisit intentions by revealing the relationship between historical storytelling and tourist perceptions from the perspective of tourists' subjective interpretations [2]. By introducing historical storytelling as an antecedent variable to enhance tourists' perception of cultural heritage tourism, and using this as a starting point, the impact of historical storytelling on tourists' revisit intention is discussed, enriching the research content of cultural heritage tourism.

Secondly, this paper further explores the chain mediation effect of destination cognitive image and perceived value as multiple mediators between historical storytelling and tourists' revisit intention. The hypothesized chain mediation effect is validated, indicating that historical storytelling significantly influence tourists' revisit intention through the chain mediation effect of destination cognitive image and perceived value. This study expands the understanding of the multiple pathway relationship between historical storytelling and tourists' revisit intention, providing alternative perspectives for cultural heritage tourism research.

Finally, many scholars currently consider place attachment as a mediating variable in different tourism contexts, confirming its positive correlation with tourists' revisit intention [2, 85]. Our study treats place attachment as a moderating variable and examines its impact on tourists' revisit intention in this research. From the empirical findings of our study, place attachment as a moderating variable only exerts a moderating effect between historical storytelling and tourists' revisit intention, while it does not significantly moderate the mediating role of

destination image and perceived value between historical storytelling and tourists' revisit intention. Therefore, this study concludes the influence of place attachment on tourists' revisit intention across different paths in the model. Additionally, this research partially addresses the deficiency in current revisit intention studies by considering moderating variables.

## Management implications

The results of this study have certain management implications for the development of the Mogao Grottoes tourism industry. It can not only guide tourism practitioners and destination managers to enhance visitor experiences and increase perceived value through specific strategies, but also provide certain references and insights for other cultural heritage sites.

The practical significance of this study mainly includes three points. First, through the results of this study, we found that historical storytelling has a significant impact on enhancing tourists' revisit intention. Therefore, in the development of cultural heritage tourism, it is important to thoroughly collect and research various local historical stories, such as religious legends, mythological stories, biographies, fables, and court life. These stories should then be cleverly combined with tourist experiences through tour guides, digital technology, and virtual reality (VR)/augmented reality (AR) techniques or more innovative means. This will showcase the destination's rich historical significance and cultural heritage to tourists, allowing them to immerse themselves in the cultural experience, enhance their satisfaction, and strive for a broader development space in the tourism market. For the Mogao Grottoes scenic area, first of all, it is necessary for the local government and the scenic area management department to establish a professional team responsible for collecting, researching, and verifying various local stories and historical legends; secondly, not only should professional training be provided to tour guides so that they can vividly and accurately tell the local historical stories, but efforts should also be made to develop the Mogao Grottoes tourism app to provide story interpretation, AR experiences, and other functions; finally, through economic incentive measures and honor selection, local residents should be encouraged to participate in the inheritance and storytelling of historical stories, in order to enhance the authenticity and interactivity of the tourist experience.

Secondly, according to this study, we found that destination cognitive image and perceived value have a significant chain-mediated effect between historical storytelling and tourists' revisit intentions. Furthermore, destination cognitive image and perceived value themselves have a significant positive impact on tourists' revisit intentions. This finding emphasizes the importance of integrating historical storytelling and destination image building in destination management. Therefore, destination managers can develop tourism products and upgrade scenic areas based on tourists' consumption preferences and purchasing behavior, aiming to enhance tourists' destination cognitive image and perceived value, thus improving the quality of scenic area services and revenue, and extending the tourism life cycle of the Dunhuang Mogao Grottoes scenic area. Specifically, the Mogao Grottoes tourist site could conduct in-depth market research to comprehend the distinct needs and preferences of various visitor segments. Based on the research outcomes, it could devise a diversified range of tourism products. For instance, targeting young tourists, innovative projects integrating technology and interactive experiences could be developed, such as Augmented Reality (AR) tours and Virtual Reality (VR) experiences, offering them an immersive and novel experience. For art and culture enthusiasts, tour packages centered around Dunhuang murals could be introduced, combining art appreciation, cultural lectures, and hands-on workshops, enabling visitors to gain a more profound understanding of Dunhuang's art heritage. At the same time, it is important to strengthen the level of scenic area services and create a high-quality tourism service

environment. By analyzing the strengths and weaknesses in the marketing and service management of the Mogao Grottoes tourism scenic area, and summarizing corresponding optimization strategies, the level of tourism scenic area service management can be improved, and its brand competitiveness can be enhanced. For example, the management department should increase funding and upgrade the infrastructure of the scenic area, such as improving restroom facilities, adding rest areas, and providing more information signs, in order to enhance the comfort of tourists during their visits. In addition, it is necessary to establish a standardized complaint handling process based on feedback from tourists, clearly defining the steps of complaint reception, recording, investigation, response, and feedback, to ensure transparency in the process and let tourists know how their complaints will be addressed.

Finally, the research results also emphasize the significant moderating role of place attachment between historical storytelling and tourists' revisit intention. This requires cultural heritage sites to pay attention to cultivating visitors' place attachment emotions in the management and development process, in order to promote visitors' tourism experience, satisfaction, and revisit intention, and thus promote the sustainable development of cultural heritage tourism. Managers should pay particular attention to showcasing the unique culture of Mogao Grottoes and its distinctive tourism products in tourists' travel experiences, allowing them to obtain a one-of-a-kind tourism experience and creating a competitive advantage through differentiation. As a representative of Dunhuang culture, Mogao Grottoes possesses abundant historical and cultural heritage resources. Integrating this unique cultural resource into tourists' travel experiences is conducive to fostering a sense of connection between people and the land, enhancing tourists' emotional attachment to the place, and thereby stimulating their revisit intention. First, the scenic area can design a series of cultural experience activities, such as Dunhuang mural copying, Silk Road history lectures, Dunhuang dance workshops, etc., to allow visitors to personally experience the unique charm of Dunhuang culture; secondly, managers can make full use of social media platforms to encourage visitors to share their experiences and stories at the Mogao Grottoes, and enhance visitors' attachment to the Mogao Grottoes through user-generated content; finally, the Mogao Grottoes can also provide personalized services for visitors, such as inviting them to attend the opening of new exhibitions and cultural lectures at the Mogao Grottoes, or offering special discounts to returning visitors, allowing visitors to feel the special attention of the Mogao Grottoes to them, effectively enhancing visitors' emotional attachment to the Mogao Grottoes and stimulating their revisit intention.

## Limitations and future research directions

The findings of this study provide relevant insights and perspectives on the relationships between historical storytelling, destination image, perceived value, place attachment, and tourists' revisit intention. However, there are still some limitations.

Firstly, in terms of research methods. The questionnaire data obtained in this study mainly relies on self-administered surveys by tourists. Given that the responses to the survey may be influenced by participants' cultural background, regional of origin, educational background, mood, cognition, and experiences, the answers provided by the respondents are inherently constrained to some extent. In future research, it could be beneficial to consider incorporating more objective data collection methods to enhance the reliability of the study. For example, using structured interviews to have face-to-face communication with tourists, delving deeper into their perspectives and opinions; or employing observational methods to collect data by directly observing tourists' behaviors and phenomena; additionally, combining big data analysis by utilizing existing datasets and data mining techniques to gather information.

Secondly, in terms of research content. This study mainly focuses on the destination cognitive image and tourists' overall perceived value, without conducting in-depth analysis on other dimensions of destination image and perceived value. The aim is to explore the influence of historical storytelling on the tourist experience and revisit intention of cultural heritage site visitors from a macro perspective. In future research, further exploration can be conducted on the emotional image and overall image within the destination image, as well as other dimensions such as transaction value and acquisition value in perceived value, in order to reveal the influence of historical storytelling on destination image and tourist perceived value from a more micro perspective. Furthermore, this study examined the moderating effect of place attachment, and found that it does not have a significant moderating effect on the mediating role of destination cognitive image and perceived value. In future research, other related moderating variables such as tourism involvement, gender, emotions, and novelty-seeking motivation can be explored.

Finally, in terms of the research subject. This study takes the famous World Cultural Heritage site, the Mogao Grottoes in Dunhuang, as a case study for empirical research. Its long history and innovative tourism experience projects are likely to further enhance the effectiveness of historical storytelling, which may limit the generalization of the research findings to other cultural heritage sites. In future research, further exploration can be conducted on the impact of historical storytelling on tourists' travel experiences and revisiting intentions at other cultural heritage sites, in order to explore potential variations in the effectiveness of these strategies.

## Supporting information

**S1 Appendix.**
(DOCX)

**S1 Data.**
(XLSX)

## Acknowledgments

Sincere gratitude is extended to the anonymous expert reviewers for their time and effort invested in the paper review process. The valuable suggestions provided by the reviewers regarding the research framework, method selection, and result analysis have greatly benefited this study.

## Author Contributions

**Conceptualization:** Qin Jianpeng.

**Data curation:** Feng Yuxin, Qin Jianpeng, Meng Weilong.

**Investigation:** Qin Jianpeng, Lv Xiaoyu, Tian Yunxia, Meng Weilong.

**Methodology:** Qin Jianpeng, Lv Xiaoyu, Tian Yunxia, Meng Weilong.

**Project administration:** Feng Yuxin.

**Resources:** Qin Jianpeng.

**Supervision:** Tian Yunxia.

**Writing – review & editing:** Feng Yuxin, Lv Xiaoyu, Tian Yunxia, Meng Weilong.

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
