## [Decision Letter · Decision Letter 0]

7 Jun 2024

PONE-D-24-19513Exploring the influence of historical storytelling on cultural heritage tourists' revisit intention: A case study of the Mogao Grottoes in DunhuangPLOS ONE

Dear Dr. qin,

Thank you for submitting your manuscript to PLOS ONE. After careful consideration, we feel that it has merit but does not fully meet PLOS ONE’s publication criteria as it currently stands. Therefore, we invite you to submit a revised version of the manuscript that addresses the points raised during the review process.

We look forward to receiving your revised manuscript.

Kind regards,

Bo Pu, Ph.D.

Academic Editor

PLOS ONE

Journal Requirements:

2. You indicated that ethical approval was not necessary for your study. We understand that the framework for ethical oversight requirements for studies of this type may differ depending on the setting and we would appreciate some further clarification regarding your research. Could you please provide further details on why your study is exempt from the need for approval and confirmation from your institutional review board or research ethics committee (e.g., in the form of a letter or email correspondence) that ethics review was not necessary for this study? Please include a copy of the correspondence as an ""Other"" file.

"National natural science foundation of China ‘Research on accurate identification of rural tourism poverty alleviation in ethical areas by combining rough set and fuzzy set’ (41661107). The phased achievement of the project ‘Research on the construction of iconic long march projects within Gansu province’ funded by the Gansu Great Wall Long March National Cultural Park Construction and Development Research Center (001053108). The 2023 graduate teaching case library construction project of Northwest Normal University: silk road China section cultural and tourism integration case library (2023YAL005)."

Additional Editor Comments:

this manuscript should be improved.

Reviewers' comments:

Reviewer's Responses to Questions

**Comments to the Author**

1. Is the manuscript technically sound, and do the data support the conclusions?

Reviewer #1: No

Reviewer #2: Yes

Reviewer #3: Yes

Reviewer #4: Yes

2. Has the statistical analysis been performed appropriately and rigorously? 

Reviewer #1: No

Reviewer #2: Yes

Reviewer #3: Yes

Reviewer #4: Yes

3. Have the authors made all data underlying the findings in their manuscript fully available?

Reviewer #1: Yes

Reviewer #2: Yes

Reviewer #3: Yes

Reviewer #4: Yes

4. Is the manuscript presented in an intelligible fashion and written in standard English?

Reviewer #1: No

Reviewer #2: No

Reviewer #3: Yes

Reviewer #4: Yes

5. Review Comments to the Author

Reviewer #1: The aim of this study is to explore the influence of historical storytelling on cultural heritage tourists' revisit intention, using the Mogao Grottoes in Dunhuang as a case study. The study examines the impact of historical storytelling on destination image, perceived value by tourists, and intention to revisit, as well as the mediating role of destination image and perceived value, and the moderating effect of place attachment. The key findings indicate that: (1) Historical storytelling significantly enhances tourists' perception of the tourism experience and intention to revisit; (2) The study supports the mediating effect of destination image and perceived value; (3) Place attachment has a significant positive moderating effect between historical storytelling and intention to revisit.

Based on the following reasons, I recommend rejecting this manuscript for publication. The study lacks a strong theoretical foundation, has methodological concerns, and does not provide sufficient depth and breadth in its exploration of the research topic.

Lack of Theoretical Grounding: The study claims to be based on the SOR (Stimulus-Organism-Response) theory, but the application of this theory is not well-justified or thoroughly integrated throughout the manuscript. The connection between the theoretical framework and the hypotheses development is not clearly articulated, raising concerns about the study's theoretical underpinnings.

Questionable Methodology: The data collection methods, including the online and offline survey distribution, are not described in sufficient detail. The sample size and composition are not adequately justified, and the potential biases or limitations of the sampling approach are not addressed. Additionally, the reliability and validity of the measurement scales used in the study are not thoroughly examined, which raises concerns about the robustness of the findings.

Narrow Scope and Limited Generalizability: The study focuses solely on the Mogao Grottoes in Dunhuang, which limits the generalizability of the findings to other cultural heritage tourism sites. The authors do not provide a clear rationale for selecting this specific case study, nor do they discuss the potential contextual factors that may have influenced the results.

Lack of Depth in Literature Review: The literature review section is relatively brief and lacks a comprehensive analysis of the existing research on historical storytelling, destination image, perceived value, place attachment, and revisit intention. The review does not adequately situate the current study within the broader context of cultural heritage tourism research, limiting the study's contribution to the field.

Insufficient Exploration of Mediating and Moderating Mechanisms: While the study examines the mediating and moderating effects, the explanations for these relationships are not well-developed. The discussion of the underlying mechanisms and the theoretical justifications for the proposed relationships are not sufficiently addressed, reducing the study's conceptual depth and the overall understanding of the complex interplay between the variables.

Incomplete Discussion of Implications: The practical implications of the study's findings are not thoroughly explored. The authors provide some general suggestions for destination managers, but the discussion lacks specific, actionable recommendations. Additionally, the study fails to address the potential limitations and future research directions, which would have strengthened the overall contribution of the manuscript.

Reviewer #2: The paper titled "Exploring the Influence of Historical Storytelling on Cultural Heritage Tourists' Revisit Intention: A Case Study of the Mogao Grottoes in Dunhuang" offers a valuable and insightful exploration into how historical storytelling can enhance cultural heritage tourism. Using the Mogao Grottoes, a UNESCO World Heritage Site, as a case study, the research applies the SOR (Stimulus-Organism-Response) theory to examine the effects of storytelling on tourists' destination image, perceived value, and revisit intentions.

One of the strengths of this paper is its clear focus on a well-defined and relevant topic. The use of historical storytelling as a means to enrich tourists' cultural perception is an innovative approach that addresses the growing need for engaging and meaningful tourist experiences. The findings that historical storytelling significantly enhances tourists' perception and intention to revisit are particularly noteworthy. This highlights the potential of storytelling as a powerful tool in cultural heritage tourism.

The research also delves into the mediating role of destination image and perceived value, as well as the moderating effect of place attachment. These insights provide a comprehensive understanding of the mechanisms through which historical storytelling influences tourist behavior. The study's support for the mediating effect of destination image and perceived value, and the positive moderating effect of place attachment, adds depth to the existing literature and offers practical implications for tourism managers.

However, the paper could benefit from the inclusion of more citations to support its claims and provide a stronger theoretical foundation. Referencing existing literature on cultural heritage tourism, storytelling, and tourist behavior would enhance the paper's academic rigor and situate its findings within the broader research context.

Additionally, the English language and overall readability of the paper could be improved. Some sections contain grammatical errors and awkward phrasing that could be polished for clarity and professionalism. A thorough revision for language accuracy and coherence would greatly enhance the paper’s readability and impact. For instance, the sentence "Visitor perception experience has always been a focal point of academic inquiry" could be revised to "Visitor perception has long been a focal point of academic inquiry." Refining the abstract and other sections for smoother flow and clearer articulation of ideas would make the paper more accessible to a broader audience.

In conclusion, "Exploring the Influence of Historical Storytelling on Cultural Heritage Tourists' Revisit Intention" is a significant and valuable contribution to the field of cultural heritage tourism. By addressing the need for more citations and improving the English language quality, the paper can further elevate its impact and utility for researchers, tourism managers, and policymakers. The innovative focus on historical storytelling as a tool for enhancing tourist engagement and promoting sustainable tourism development is particularly noteworthy, providing a solid foundation for future research and practical applications in the tourism industry.

Recommended article for citations:

https://journals.economic-research.pl/eq/article/view/2398/2086

and

https://www.tandfonline.com/doi/full/10.1080/14616688.2023.2224043

and

https://www.tandfonline.com/doi/full/10.1080/01490400.2021.1908193

Etc.

Reviewer #3: In the research, the mediating effect of destination image and perceived value on the effect of historical storytelling on revisit intention was determined. On the other hand, the regulatory role of place attachment in the effect of historical storytelling, destination image and perceived value on revisit intention has been tried to be explained. The following findings were made in the research;

1. The model and hypotheses of the research are explained with the support of the literature.

2. The minimum number was provided in the sample selection of the research, but since it was a multivariate research, a sample over 384 would have been more appropriate. It is recommended that sampling adequacy be explained by power analysis or another method.

3.The research findings are quite meaningful. The results are supported by the literature, but the striking results can be emphasized a little more. For example, since historical storytelling is an important concept, it should be supported with concrete storytelling.

4. Particular emphasis should be placed on the regulatory role of place attachment, with literature support.

5. The purpose of the research, model, hypotheses and analyzes are compatible with each other.

I wish you good luck.

Reviewer #4: 1.The author needs to revise the abstract, and the first part briefly writes down the background and purpose.

2.Add a reference to the first paragraph of the introduction regarding the number of cultural heritages worldwide. Where does the first sentence of the second paragraph of the introduction come from?

3.Please note the full name of the acronym SOR theory after its first appearance. It is suggested that the third and fourth paragraphs of the introduction be merged, followed by a condensed version of the introduction to make this part as concise as possible.

4.It is recommended to compress the content of the literature review in Chapter II. The literature review should be closely centered on the topic of the paper. The relationship between the literature review of several sections is briefly introduced in the first paragraph of chapter 2.

6. PLOS authors have the option to publish the peer review history of their article (what does this mean?). If published, this will include your full peer review and any attached files.

Reviewer #1: No

Reviewer #2: No

Reviewer #3: No

Reviewer #4: No

---

## [Author Response · Author response to Decision Letter 0]

1 Jul 2024

Dear editors and reviewers of PLOS ONE:

Thank you for giving us the opportunity to submit a revised draft of the manuscript ‘Exploring the influence of historical storytelling on cultural heritage tourists' revisit intention: A case study of the Mogao Grottoes in Dunhuang’ for publication in the Journal of PLOS ONE. We appreciate the time and effort that you and the reviewers dedicated to providing feedback on our manuscript and are grateful for the insightful comments on and valuable improvements to our paper. We have incorporated most of the suggestions made by the reviewers. Those changes are highlighted in the manuscript. Please see below, in blue, for a point-by-point response to the reviewers’ comments and concerns. All page numbers refer to the revised manuscript file with tracked changes. For more information, please see the Response to Reviewers.

---

## [Decision Letter · Decision Letter 1]

15 Jul 2024

Exploring the influence of historical storytelling on cultural heritage tourists' revisit intention: A case study of the Mogao Grottoes in Dunhuang

PONE-D-24-19513R1

Dear Dr. qin,

We’re pleased to inform you that your manuscript has been judged scientifically suitable for publication and will be formally accepted for publication once it meets all outstanding technical requirements.

Kind regards,

Bo Pu, Ph.D.

Academic Editor

PLOS ONE

Additional Editor Comments (optional):

Thanks for your hardwork on this manuscript.

Reviewers' comments:

Reviewer's Responses to Questions

**Comments to the Author**

1. If the authors have adequately addressed your comments raised in a previous round of review and you feel that this manuscript is now acceptable for publication, you may indicate that here to bypass the “Comments to the Author” section, enter your conflict of interest statement in the “Confidential to Editor” section, and submit your "Accept" recommendation.

Reviewer #3: All comments have been addressed

Reviewer #4: All comments have been addressed

2. Is the manuscript technically sound, and do the data support the conclusions?

Reviewer #3: Yes

Reviewer #4: Partly

3. Has the statistical analysis been performed appropriately and rigorously? 

Reviewer #3: Yes

Reviewer #4: Yes

4. Have the authors made all data underlying the findings in their manuscript fully available?

Reviewer #3: Yes

Reviewer #4: Yes

5. Is the manuscript presented in an intelligible fashion and written in standard English?

Reviewer #3: Yes

Reviewer #4: Yes

6. Review Comments to the Author

Reviewer #3: Dear Author/s

Thank you for your kind response to my revision suggestions. Your answers are self-explanatory enough.

Good luck.

Reviewer #4: The author basically modified the questions I raised and suggested that the English expression of the full text be refined.

7. PLOS authors have the option to publish the peer review history of their article (what does this mean?). If published, this will include your full peer review and any attached files.

Reviewer #3: No

Reviewer #4: No

---

## [Editor Report · Acceptance letter]

11 Sep 2024

PONE-D-24-19513R1 

PLOS ONE

Dear Dr. Jianpeng, 

I'm pleased to inform you that your manuscript has been deemed suitable for publication in PLOS ONE. Congratulations! Your manuscript is now being handed over to our production team.

Kind regards, 

on behalf of

Dr. Bo Pu 

Academic Editor

PLOS ONE